# The Role of Multimodal Imaging in Pathological Response Prediction of Locally Advanced Cervical Cancer Patients Treated by Chemoradiation Therapy Followed by Radical Surgery

**DOI:** 10.3390/cancers15123071

**Published:** 2023-06-06

**Authors:** Tina Pasciuto, Francesca Moro, Angela Collarino, Maria Antonietta Gambacorta, Gian Franco Zannoni, Marco Oradei, Maria Gabriella Ferrandina, Benedetta Gui, Antonia Carla Testa, Vittoria Rufini

**Affiliations:** 1Data Collection G-STeP Research Core Facility, Fondazione Policlinico Universitario A. Gemelli IRCCS, 00168 Roma, Italy; 2Gynecologic Oncology Unit, Department of Woman and Child Health and Public Health, Fondazione Policlinico Universitario A. Gemelli IRCCS, 00168 Roma, Italy; 3Nuclear Medicine Unit, Fondazione Policlinico Universitario A. Gemelli IRCCS, 00168 Roma, Italy; angela.collarino@policlinicogemelli.it (A.C.);; 4Radiation Oncology Unit, Fondazione Policlinico Universitario A. Gemelli IRCCS, 00168 Roma, Italy; 5Section of Radiology, University Department of Radiological Sciences and Hematology, Università Cattolica del Sacro Cuore, 00168 Roma, Italy; 6Gynecopathology Unit, Department of Woman and Child Health and Public Health, Fondazione Policlinico Universitario A. Gemelli IRCCS, 00168 Roma, Italy; 7Section of Pathology, Department of Woman and Child Health and Public Health, Università Cattolica del Sacro Cuore, 00168 Roma, Italy; 8ALTEMS (Graduate School of Health Economics and Management), Università Cattolica del Sacro Cuore, 00168 Roma, Italy; 9Section of Obstetrics and Gynecology, University Department of Life Sciences and Public Health, Università Cattolica del Sacro Cuore, 00168 Roma, Italy; 10Radiology Unit, Fondazione Policlinico Universitario A. Gemelli IRCCS, 00168 Roma, Italy; 11Section of Nuclear Medicine, University Department of Radiological Sciences and Hematology, Università Cattolica del Sacro Cuore, 00168 Roma, Italy

**Keywords:** cervical cancer, chemoradiation, ultrasound, magnetic resonance imaging, ^18^F-FDG-PET/CT, pathological response prediction

## Abstract

**Simple Summary:**

In patients with locally advanced cervical cancer, the availability of imaging techniques for accurately defining the residual tumor would be clinically relevant for selecting patients who could be offered a more tailored surgery. The novelty of this prospective study is the development of multiparametric predictive models of histopathological response using a unique data set with three imaging modalities (transvaginal ultrasound, magnetic resonance (MRI) and ^18^F-FDG-PET/CT) evaluated at three time points (“baseline”, two (“early”) and five (“final”) weeks after treatment). In a cohort of 88 patients, the predictive models retrieved integrating morphometric, vascular, perfusion and metabolic parameters, demonstrated that two imaging approaches (MRI and PET/CT at “final” evaluation or PET/CT at “baseline” and “final” evaluation) are sufficient to identify possible residual disease after chemotherapy. These findings could be useful in selecting patients with residual disease, helping clinicians to tailor the radicality of the surgical approach.

**Abstract:**

Purpose: This study aimed to develop predictive models for pathological residual disease after neoadjuvant chemoradiation (CRT) in locally advanced cervical cancer (LACC) by integrating parameters derived from transvaginal ultrasound, MRI and PET/CT imaging at different time points and time intervals. Methods: Patients with histologically proven LACC, stage IB2–IVA, were prospectively enrolled. For each patient, the three examinations were performed before, 2 and 5 weeks after treatment (“baseline”, “early” and “final”, respectively). Multivariable logistic regression models to predict complete vs. partial pathological response (pR) were developed and a cost analysis was performed. Results: Between October 2010 and June 2014, 88 patients were included. Complete or partial pR was found in 45.5% and 54.5% of patients, respectively. The two most clinically useful models in pR prediction were (1) using percentage variation of SUV_max_ retrieved at PET/CT “baseline” and “final” examination, and (2) including high DWI signal intensity (SI) plus, ADC, and SUV_max_ collected at “final” evaluation (area under the curve (95% Confidence Interval): 0.80 (0.71–0.90) and 0.81 (0.72–0.90), respectively). Conclusion: The percentage variation in SUV_max_ in the time interval before and after completing neoadjuvant CRT, as well as DWI SI plus ADC and SUV_max_ obtained after completing neoadjuvant CRT, could be used to predict residual cervical cancer in LACC patients. From a cost point of view, the use of MRI and PET/CT is preferable.

## 1. Introduction

The standard treatment of locally advanced cervical cancer (LACC) is exclusive chemoradiation therapy (CRT) [1,2]. According to a Phase III randomized study, an alternative strategy is neoadjuvant CRT followed by radical surgery [3,4,5,6,7]. This approach, which gave similar results in terms of response compared with exclusive CRT, provides prognostic information as patients reaching a pathological complete response after neoadjuvant CRT show better disease-free survival and longer overall survival than those achieving partial response [3,8,9]. In this setting, the identification of a noninvasive biomarker of partial response after neoadjuvant CRT in LACC patients is an important clinical issue. The availability of imaging techniques able to accurately define the residual tumor, would be clinically relevant for selecting patients who could be spared surgery or at least be offered a more tailored surgery.

From this perspective, we performed a prospective study with the aim to analyze the predictive ability of transvaginal ultrasound examination (TUS), magnetic resonance imaging (MRI), positron emission tomography/computed tomography (PET/CT), as well as their complementary role in detecting residual disease after neoadjuvant CRT. In previous studies, we separately explored several single quantitative or semi-quantitative parameters of each individual imaging method, namely, TUS vascular indices, TUS contrast and morphological parameters, MRI tumor volume, MRI diffusion-weighted imaging signal intensity (DWI SI) and mean apparent diffusion coefficient (ADC_mean_), as well as ^18^F-FDG-PET/CT parameters such as maximum standardized uptake value (SUV_max_), SUV_mean_, metabolic tumor volume (MTV), and total lesion glycolysis (TLG) [10,11,12,13,14]. We showed that before, during and after neoadjuvant CRT, some parameters were significantly different in patients with residual disease at histopathology (partial responders) compared with those with no residual disease (complete responders). However, no one parameter alone provided a high level of diagnostic performance.

This study aimed to develop multiparametric predictive models for residual disease after neoadjuvant CRT in LACC patients by integrating morphometric, vascular, perfusion and metabolic parameters derived from three imaging methods (TUS, MRI and ^18^F-FDG-PET/CT) obtained at different time points and time intervals.

## 2. Materials and Methods

This prospective study was approved by the Ethics Committee of Fondazione Policlinico Universitario Agostino Gemelli IRCCS–Università Cattolica del Sacro Cuore (ID P/572/CE/2010). All the subjects signed an informed consent form. All efforts were made to avoid selection bias, and consecutive eligible patients with histologically proven LACC (any histology) and Stage IB2–IVA disease (according to the International Federation of Gynecology and Obstetrics (FIGO) classification 2009 [15]) were enrolled at the Gynecologic Oncology Unit. Other inclusion and exclusion criteria have been previously reported [10]. Neoadjuvant CRT included whole pelvic irradiation (1.8 cGy/fraction, 22 fractions), with a total dose of 39.6 Gy, and an additional dose of 10.8 Gy to the primary tumor and parametrium through the concomitant boost technique (0.9 cGy/fraction, 12 fractions every other day) [9]. Concomitant chemotherapy included cisplatin (20 mg/m^2^, 2 h intravenous infusion) during the first 4 and the last 4 days of treatment and capecitabine (1300 mg/m^2^/daily, orally) during the first 2 and last 2 weeks of treatment. Patients were evaluated according to the Response Evaluation Criteria for Solid Tumors (RECIST) 4–6 weeks after completion of CRT [16]. In patients achieving response, radical hysterectomy and pelvic (with or without aortic) lymphadenectomy were planned within 6–8 weeks from completion of CRT. Patients showing no change or disease progression at MRI and PET/CT were treated with salvage chemotherapy.

All three imaging techniques were performed approximately 3 weeks before treatment (“baseline”), after 2 weeks of treatment (“early” evaluation), and at 5 weeks after the end of treatment (“final” evaluation). The planned time-interval including the three imaging techniques was usually 3 days and did not exceed one week in any case. The percentage variation (delta) in the TUS, MRI and PET/CT parameters were evaluated for the “baseline”–“early” and “baseline”–“final” evaluations. Tumor volumes at TUS and MRI were calculated with the ellipsoid formula (antero-posterior × cranio-caudal × latero-lateral diameter × π/6). Imaging analysis of each modality was performed blind to the others and to the histopathology.

According to the interpretation criteria described below for each methodology, any abnormality in the cervix was interpreted as a tumoral lesion, which was subsequently correlated with the histopathology in each case.

The histopathological evaluation was performed by a skilled gynecologic oncologist pathologist (G.F.Z.). At pathology, cervical residual disease was defined as: absent (complete response, pR0); microscopic (presence of tumor foci <3 mm, pR1); and macroscopic (presence of tumor foci ≥3 mm, pR2) [17]. The results obtained with the three imaging modalities were compared with those of the histopathology.

### 2.1. Ultrasound Methodology and Data Analysis

To avoid interobserver variability, all ultrasound examinations were performed by the same examiner (A.C.T.), who has more than 15 years of experience in gynecologic ultrasound. The tumor characteristics were assessed with standardized techniques including 2D and 3D grayscale and power/color Doppler examination of cervical tumor volumes, and contrast-enhanced examination with infusion of SonoVue contrast agent (Bracco Imaging SpA, Milan, Italy) [10]. A subjective semi-quantitative assessment of the amount of detectable blood flow was made using the color score, as previously described [18]. 3D power Doppler indices included vascularization index (VI), flow index (FI) and vascularization flow index (VFI). The contrast-enhanced ultrasound examination was performed using CnTI™ (contrast-tuned imaging) technology (Esaote) integral to the transvaginal probe and with the ultrasound contrast agent SonoVue, as described previously [10]. The bolus model considering the wash-in/wash-out kinetics was used for the analysis. Perfusion parameters, such as wash-in rate, peak enhancement, rise time and area under the time–intensity curves during wash-in and wash-out were calculated in a specific region of interest corresponding to the residual tumor detected within the cervix and to the whole cervix. All regions of interest were drawn by a single operator (T.P.) on the largest diameter of the residual lesion identified by the ultrasound examiner (F.M.). The regions of interest were analyzed using the software package VueBox^®^ 6.0 (vuebox.bracco.ch/php/Support.php accessed on 1 September 2015, Bracco Imaging SpA, Milan, Italy).

### 2.2. MRI Methodology and Data Analysis

Pelvic conventional and DW-MRI were performed and reviewed according to a previously described protocol using a 1.5-T superconducting magnet (Echospeed Horizon and Infinity, GE Medical Systems, Milwaukee, WI, USA) [12]. Cervical tumor diameters, volume and ADC_mean_ values were measured at “baseline”, “early” and “final” examination. Tumor diameters were assessed on axial and sagittal FSE T2-WI. The maximum tumor diameters (maxTD) were recorded in the three dimensions obtained in the sagittal and in the axial T2-WI. DWI images were analyzed qualitatively, referring to signal intensity of the tumor, which was classified as hyperintense or hypointense in comparison with the adjacent skeletal muscle. The ADC map was generated by using a designated workstation (Horizon Advantage GE Medical System or Advanced Workstation; GE Medical Systems) and was analyzed using the Functool dynamic analysis tool (GE Medical Systems). Three freehand regions of interest (ROIs) were drawn on a single DW image where the lesion diameters were maximum, using axial T2-WI as guidance. Areas of necrosis within the tumor were avoided. The ROIs were copied to the corresponding ADC map, and the mean ADC (ADC_mean_) was obtained. In the absence of high signal intensity on DWI, the ROI was placed on the cervical stroma, in the site of the tumor at the baseline DW-MRI. Tumor response was classified as follows: (1) complete response in patients with total restoration of the zonal anatomy of the cervix (i.e., demonstration of homogeneously hypointense stroma on T2-WI) or with areas of intermediate or high signal intensity on T2-WI and no signal intensity on DWI or high signal intensity on DWI but with an ADC_mean_ value > 1.1 × 10^−3^ mm^2^/s, and (2) partial response in patients with residual disease based on evidence of a residual hyperintense mass within the cervix on T2-WI with evidence of signal intensity on DWI and an ADC_mean_ value < 1.1 × 10^−3^ mm^2^/s [19]. According to these criteria, a dichotomous MRI parameter high DWI SI plus ADC was defined and set as equal to 1 in cases of high DWI plus ADC_mean_ ≤ 1.1 × 10^−3^ mm^2^/s, and equal to 0 in cases of high DWI and ADC_mean_ >1.1 × 10^−3^ mm^2^/s or in all cases with low DWI SI.

### 2.3. PET/CT Methodology and Data Analysis

Standard PET/CT scans (without iodine contrast) were acquired from the skull to pelvis according to a previously described protocol using 3D Gemini GXL Philips Medical Systems at 60 min (±10 min) after ^18^F-FDG injection (3 MBq/kg) and reconstructed using the line-of-response row-action maximum likelihood algorithm (three iterations and 33 subsets, voxel size: 4 × 4 × 4 mm^3^) [14]. The images were reviewed on Siemens Healthcare Syngo.via workstations. The volumes of interest (VOI) were carefully placed in the same anatomic site on all three PET scans for each patient; care was taken not to include bladder activity in the VOI. SUV_max_, SUV_mean_, MTV and TLG were calculated using a gradient-based method (PET Edge tool of MIM Encore software, version 6.9.3; MIM Software Inc., Cleveland, OH, USA) [14,20]. Any focus of ^18^F-FDG uptake at the primary site higher than the surrounding background was considered abnormal and interpreted as positive. Tumor response was classified as follows: (1) complete response in patients with absence of abnormal ^18^F-FDG uptake at the site of the cervical tumor; (2) partial response in patients with residual abnormal ^18^F-FDG uptake at this site [14].

### 2.4. Statistical Analysis

Sample size was calculated to detect a 15% difference in the accuracy of MRI. Based on MRI diagnostic accuracy = 75% (p0), type-1 error = 0.01, and type-II error = 0.1 (power, 90%), a total of 86 patients would be required. Assuming a dropout rate of around 10%, a final sample size of 95 patients was planned [10].

Clinical, pathological, and imaging characteristics were described as n (%) or median (min-max) as appropriate. For the analysis, patients were divided in two groups (complete vs. partial) according to pathological response (reference standard).

The Shapiro–Wilk test was used to test normality and comparisons between the two groups were made with the Mann–Whitney U test and χ2 test as appropriate. Imaging parameters were analyzed at “baseline”, “early”, and “final” time points. Moreover, differences between “baseline”–“early”, and “baseline”–“final” time intervals were also evaluated according to the following formula:(1)$$\left(100\times [\left(“\mathrm{Baseline}”\;\mathrm{Value}-“\mathrm{Early}”\;\mathrm{or}\;“\mathrm{Final}”\;\mathrm{Value}\right)/“\mathrm{Baseline}”\;\mathrm{Value}]\right)$$

In the present study, only a selection of parameters described in previous studies [10,11,12,13,14] with a *p* value less than 0.05 when comparing partial versus complete pathological responders at inferential analysis were analyzed (Table 1).

At each time point (“baseline”, “early” and “final”) or time interval (“baseline”–“early” and “baseline”–“final”), the selected parameters were included in univariable logistic regression analyses in order to evaluate their performance in pathological response prediction. Those parameters that showed a *p* value less than 0.05 were included in multivariable logistic regression models using the stepwise backward method. The significance level for removal from the model was set at 0.1 and the method was chosen according to the sample size, as suggested in the literature [21]. Multivariable models were developed in order to evaluate the performance of the multiparametric pathological response prediction according to two criteria: The first aimed to analyze the strength of each single imaging method alone (TUS, MRI, PET/CT), joining parameters detected by the same imaging. The second evaluated the strength of combining parameters detected by different imaging methods.

To avoid collinearity, for each examination, only the parameter with the lowest *p* value in the univariable analysis was included in the multivariable analysis, both for morphometric (maximum tumor diameter and tumor volume) and SUV (SUV_max_ and SUV_mean_) parameters. In the case of equal *p* values, the criterion used for parameter selection was the maximization of area under the curve (AUC).

All estimations and AUC values were provided with 95% confidence intervals (CIs). AUC values between 0.70 and 0.80 were considered acceptable, those between 0.81–0.90 excellent, and those > 0.9 outstanding [22]. When there were superimposable results, the most clinically useful model was selected to maximize the lower 95% CI limit.

A two-sided test was used and a *p* value < 0.05 was considered statistically significant. No imputation was carried out for missing data. The statistical analysis was performed by an experienced biostatistician (TP) using STATA software (STATA/BE 17.0 for Windows, StataCorp LP, College Station, TX 77845, USA).

### 2.5. Cost Analysis

An analysis of the costs related to the different models developed was carried out.

Each examination cost was valued according to the outpatient tariff of the Lazio Region (similar to that of the other Italian Regions) which represents the reimbursement that the Regional Health System recognizes to the structures that perform the services. The examinations were identified according their specific regional codes.

## 3. Results

Between October 2010 and June 2014, 108 patients were initially screened. Of these, 16 refused early evaluation and two died during CRT; 90 patients completed neoadjuvant CRT and imaging studies, two of whom showed progressive disease at the final assessment and were excluded. Thus, 88 patients were included in the final analysis (Figure 1): 11 patients with adenocarcinoma (12.5%), and 77 patients with squamous cell carcinoma (87.5%).

The clinical and pathological features of the study population are summarized in Table 2.

Overall, 40/88 (45.5%) had pR0, while 48/88 (54.5%) patients had PR, including 21 pR1 (23.8%) and 27 pR2 (30.7%). A significant difference was found for grading of differentiation and metastatic LNs at histology, whereas borderline significance was found for histotypes. At histopathology, metastatic pelvic lymph nodes were detected in 10/88 (11.4%) patients and in all patients with a residual cervical tumor.

Of 95 parameters extracted from the three imaging modalities, only 34 (36%) were eventually considered in the present study (Table 1).

Appendix A summarizes the TUS, MRI and PET/CT parameters that significantly differed between patients with a partial response and those with a complete response at “baseline”, “early” and “final” examinations as absolute values and their percentage delta variations (Δ) “baseline”–“early” and “baseline”–“final”. These 34 diagnostic parameters were considered for both uni- and multivariable analyses as appropriate.

Table 3 shows uni- and multivariable analysis of combined parameters from the same imaging to predict the pathological partial response at each time point or time interval. All multivariable models had an acceptable AUC ranging from 0.70 to 0.80 with a lower limit of 95% CI ranging from 0.57 to 0.71. In synthesis, the parameters with an independent predictive role within the same imaging method in the multivariable analysis were:At “baseline”: color score and tumor peak enhancement for TUS, none for MRI (none of these parameters was analyzed in the present study), and SUV_mean_ for PET/CT;At “early” examination: maxTD and VI for TUS, maxTD for MRI, and MTV for PET/CT;At “final” examination: no parameters for TUS, maxTD and the combined parameter of high DWI SI plus ADC for MRI, and SUV_max_ for PET/CT;For Δ “baseline”–“early” parameters: ΔTumor volume (%) for TUS, ΔTumor volume (%) for MRI, ΔSUV_mean_ (%), ΔMTV (%), and ΔTLG (%) for PET/CT;For Δ “baseline”–“final” parameters: none for TUS (parameters not evaluated); Δmaximum tumor diameter and ΔADC_mean_ (%) for MRI, ΔSUV_max_ (%) for PET/CT.

The model with the highest lower 95% CI limit was that considering the variation of SUV_max_ values evaluated at “baseline” and “final” evaluation (AUC: 0.80, 95% CI: 0.71–0.90).

Table 4 shows the results of the multivariable analysis when the predictive parameters of the three imaging methods are combined. In summary, five models with statistically significant parameters were identified:Model 1, at “baseline” examination: VFI and SUV_mean_;Model 2, at “early” examination: only one ultrasound parameter (vascularization index);Model 3, at “final” examination: high DWI SI plus ADC and SUV_max_;Model 4, for Δ “baseline”–“early” parameters: ΔSUV_mean_ (%), ΔMTV (%), ΔTLG (%);Model 5, for Δ “baseline”–“final” parameters: ΔSUV_max_ (%).

All the models had an acceptable AUC with a superimposable 95% CI (lower 95% CI limit ranging from 0.61 to 0.73. Model 3 (at “final” examination) and model 5 (Δ “baseline”–“final”) had an excellent AUC showing the highest lower 95% CI limit ≥ 0.72. Moreover, the results for model 5 were similar to the ones shown in Table 3, and slight differences are only apparent due to the different samples used for the model development (82 patients for model 5 and 88 patients for the model shown in Table 3). According to the results of the logistic regression at “final” examination, the probability ($y_{t}$) of the subject having a partial pathological response can be determined by:(2)$$y_{t}=\frac{e^{\left(-3.49+1.40{*x}_{i1}+0.91{*x}_{i2}\right)}}{1+e^{\left(-3.49+1.40{*x}_{i1}+0.91{*x}_{i2}\right)}}\;i=1\ldots 82$$
$$\left(x_{i1};x_{i2}\right)=\left({\mathrm{Evaluation}\;\mathrm{according}\;\mathrm{to}\;\mathrm{high}\;\mathrm{DWI}\;\mathrm{SI}\;\mathrm{plus}\;\mathrm{ADC}\;}_{i};{\mathrm{SUV}\;\max\;}_{i}\right)$$

The parameter high DWI SI plus ADC (x_i1_ for i = 1…82) was a dichotomous parameter and SUV_max_ (x_i2_ for i = 1…82) was a continuous parameter (Figure 2).

For instance, when applying this model, a patient with high DWI SI plus ADC = 0 and SUV_max_ = 1.26 at “final” examination, would have an 8.8% probability of having residual disease (or a pathological response). Conversely, a patient with high DWI SI plus ADC = 1 and SUV_max_ = 10.5 at “final” examination would have a 99.9% probability of having residual disease.

Table 5 shows the results of the cost analysis.

## 4. Discussion

Using multivariable analysis, this prospective study generated models including the best parameters of three imaging methods (TUS, MRI and PET/CT) for predicting partial pathological response after neoadjuvant CRT in LACC patients. We found that the use of multiple parameters retrieved from the same imaging method resulted in models with superimposable acceptable AUCs for before, during and after treatment. The model considering ΔSUV_max_ values evaluated for the time interval from “baseline” to “final” evaluation was considered the most clinically useful. The use of parameters derived from the three imaging methods showed similar results in terms of superimposable AUC for most of the evaluations and confirmed the role of ΔSUV_max_ in pathological response prediction for the time interval “baseline”–“final” evaluation. Moreover, another model considering high DWI SI plus ADC of MRI and SUV_max_ of PET/CT at “final” examination (model 3) had an excellent AUC: patients with both high DWI with ADC_mean_ ≤ 1.1 × 10^−3^ mm^2^/s and high SUV_max_ detected 5 weeks after treatment are more likely to have a partial pathological response.

Regarding cost analysis, model 1 and 2, although less expensive, cannot be taken into consideration as they include ultrasonography plus color Doppler which are not considered in the main national and international clinical guidelines. Model 3, which includes both MRI and PET/CT seems the most appropriate as it provides the necessary clinical information at a lower cost than models 4 and 5, which include PET/CT at two time points. Furthermore, in the Italian context, adoption of model 3 would allow lightening the workload for PET centers, which have less availability in the national health system. A further possible advantage, for facilities equipped with an integrated MRI/PET device, is performing both exams in a single session, resulting in a more timely and comprehensive report and a single outpatient access.

To our knowledge, this is the first study elaborating a predictive model of pathological partial response including the parameters of three different diagnostic methods. Indeed, other studies investigated the role of TUS, MRI and PET/CT parameters in LACC patients, but none of them evaluated the complementary role of the three methods before, during and after CRT [23,24,25,26]. Only a few studies assessed the role of ultrasound parameters in predicting residual tumor, with inconsistent results [23,24,27,28,29]. Among studies assessing the role of MRI, the most important is a retrospective study that investigated the performance of DWI-MRI in detecting pathologically residual disease after CRT in 52 cervical cancer patients. The authors reported values of sensitivity, specificity, and accuracy for high DW signal intensity of 65%, 63%, 65%, respectively, and for low ADC (visual), values of 35%, 90%, 69%, respectively [25]. Another two studies investigated the predictive role of PET/CT parameters for response to CRT but both included a small number of cases (24 cases and 34 cases, respectively) and assessed only clinical response without examining the histological data [26,30]. In our study, volumetric-based metabolic parameters such as MTV and TLG, which have been widely studied in the recent literature, did not show an optimal performance or clinical usefulness (significant values only at “early” imaging) [31,32,33].

We chose to perform “early” imaging after two weeks of treatment considering that at this time, radiation-induced inflammation was supposed to be very low. We chose to perform “final” imaging 5 weeks after completion of neoadjuvant CRT, a time that was earlier than standard of care, which is 3 months after completion of exclusive CRT. In fact, it is assumed that at this time (3 months from the end of CRT), radio-induced inflammation should have resolved, with few false positive results. It must be stressed that the total RT dose used in our protocol was lower than that used with exclusive CRT as surgery substituted for utero-vaginal brachytherapy. In any case, the time we chose was a satisfactory compromise, considering the time of surgery, which was planned within 8 weeks from the end of CRT [14].

A potential limit of the study is that we excluded from the analysis patients with no response at neoadjuvant CRT; however, the number of patients with no response was too small to be included (two patients). Second, we are aware that a more recent FIGO stage (2018) is available [34,35], which increases the role of diagnostic techniques by using imaging findings for tumor size measurement (which is a prerogative of MRI) and lymph node assessment (which is a prerogative of PET/CT) [36]. In any case, according to the prospective design of the study, we decided to adopt the previous FIGO stage classification as the inclusion period time was 2010–2014. We are aware that the involvement of only one examiner, which was applied only for ultrasonography, introduces some uncertainty into the data. However, when planning the study, we decided to involve one single ultrasound examiner with very high experience due to the specialized process of real-time image selection and interpretation, as well as the additional use of complex diagnostic techniques such as infusion of SonoVue. Moreover, our study may have limited implications for clinical practice; indeed, the treatment performed in our protocol (neoadjuvant treatment followed by radical surgery) is not a widespread strategy in the world. However, in this way we built a predictive model according to a very strong gold standard for a pathology. Additionally, having a predictive model for cervical residual disease after chemoradiation may be useful for customizing treatment to minimize dosage and side-effect risks, even in patients undergoing exclusive CRT. For example, in patients with evidence of no residual disease after radiotherapy, the radiotherapist may decide to use a low dose of brachytherapy to minimize the risk of fistula. Finally, we are aware that the current patient population entirely overlap with our previous studies, which using the same dataset, separately analyzed the role of the three imaging techniques in single parameter prediction of histopathological response after neoadjuvant CRT in LACC patients [10,11,12,13,14]. In fact, the large number of investigated parameters required a skimming process to select those parameters most representative for each imaging examination. The originality of this study is in the attempt to merge the results obtained from the evaluation of more than one parameter to improve the predictive performances either using a single imaging method or integrating more than one. In any case, the current manuscript differs in analytic methods and provides new, additional analyses already planned in the original protocol, which are complementary to those of previous studies.

## 5. Conclusions

The novelty of this study is the development of multiparametric predictive models using a unique data set with three imaging modalities, three time points, histopathological correlations, as well as a thorough consideration of different imaging parameters. Moreover, the present study provided some new findings. First, imaging performed after two weeks of treatment (“early” examination) is not so advantageous in clinical practice, regardless of the imaging procedure. Second, the predictive models demonstrated that two imaging approaches (MRI and PET/CT at “final” evaluation or PET/CT at “baseline” and “final” evaluation) are sufficient to identify possible residual disease after CRT. These findings could be useful in counselling patients before treatment, and above all, in selecting patients with residual disease, thereby helping clinicians to tailor the radicality of the surgical approach. From a cost point of view, the use of MRI and PET/CT at “final” evaluation is preferable.

## Figures and Tables

**Figure 1 cancers-15-03071-f001:**
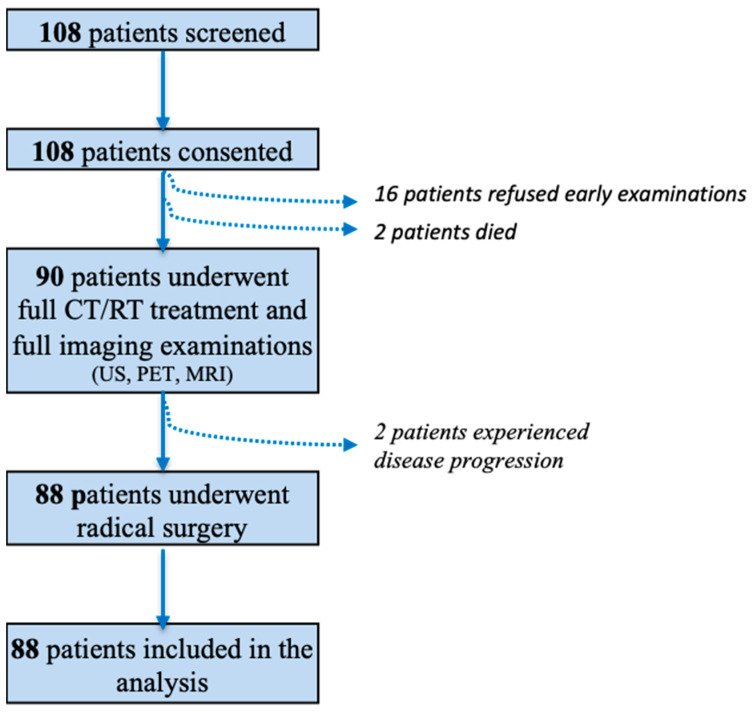
Flow-chart of the study population.

**Figure 2 cancers-15-03071-f002:**
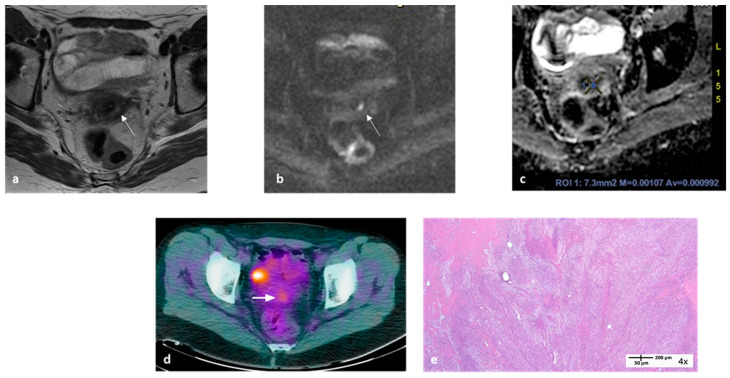
MRI and ^18^F-FDG-PET/CT images at “final” examination of a 50-year-old woman with locally advanced cervical cancer. Transaxial MRI showed a small residual hyperintense area within the cervix on T2-WI ((**a**), arrow) with evidence of high signal intensity on DWI ((**b**), arrow) and an ADC_mean_ value ≤ 1.1 × 10^−3^ mm^2^/s (**c**). Transaxial PET/CT image showed an area of focal uptake within the cervix with SUV_max_ 3.3 ((**d**), arrow); the area of focal intense uptake is due to bladder activity. Applying the formula reported in the text, the probability (y_t_) of this patient having a partial pathological response was 71.4%. Histopathology showed macroscopic residual disease (**e**).

**Table 1 cancers-15-03071-t001:** List of the imaging parameters collected for the whole study highlighting those included in the logistic regression analysis performed in the present study.

Characteristic *	“Baseline” Examination	“Early” Examination	“Final” Examination	Δ “Baseline”–“Early” Evaluation	Δ “Baseline”–“Final” Evaluation
TUS					
Tumor volume					evaluation not performed
Maximum tumor diameter				
Echogenicity				not applicable
Color score				not applicable
Vascular indices				
VI				
FI				
VFI				
3D tumor volume				
Tumor peak enhancement				
Tumor rise time				
Tumor wash-in rate				
Tumor wash-in				
Tumor wash-out				
MRI					
Tumor volume					
Maximum tumor diameter					
Intensity				not applicable	not applicable
High DWI SI				not applicable	not applicable
High DWI SI plus ADC_mean_ ≤ 1.1 × 10^−3^ mm^2^/s or ΔADC_mean_					
PET/CT					
SUV_max_					
SUV_mean_					
MTV					
TLG					

TUS: transvaginal ultrasound. VI: vascularization index. VFI: vascularization flow index. MRI: magnetic resonance imaging. PET/CT: positron emission tomography/computer tomography. SUV: standardized uptake value. MTV: metabolic tumor volume. TLG: total lesion glycolysis. DWI SI: diffusion weighted imaging signal intensity. ADC: apparent diffusion coefficient. * For the evaluation of the time intervals Δ “baseline”–“early” and Δ “baseline”–“final”, the percentage variation in the continuous parameters was evaluated according to Formula (1). 

: parameter not statistically different between patients with pathological complete response and patients with pathological partial response; not evaluated in the present study. 

: parameter statistically different between patients with pathological complete response and patients with pathological partial response; evaluated in the present study.

**Table 2 cancers-15-03071-t002:** Clinical and pathological characteristics of the study population.

Characteristics	All Cases n = 88	Partial Response * n = 48	Complete Response n = 40	*p* Value
Age (years)	49.5 (22–75)	49 (22–75)	50 (31–72)	0.893
FIGO stage				0.872
I B2	3 (3.4)	2 (4.2)	1 (2.5)	
II A	9 (10.2)	5 (10.4)	4 (10.0)	
II B	63 (71.6)	34 (70.8)	29 (72.5)	
III A	4 (4.5)	3 (6.3)	1 (2.5)	
III B	9 (10.2)	4 (8.3)	5 (12.5)	
Pelvic lymph node involvement at imaging	40 (45.5)	21 (43.8)	19 (47.5)	0.725
Grading of differentiation at staging †				**0.026**
G1	2/79 (2.5)	0/43 (0)	2/36 (5.6)	
G2	56/79 (70.9)	27/43 (62.8)	29/36 (80.6)	
G3	21/79 (26.6)	16/43 (37.2)	5/36 (13.9)	
Histotype				0.052
Adenocarcinoma	11 (12.5)	9 (18.8)	2 (5.0)	
Squamous	77 (87.5)	39 (81.2)	38 (95.0)	
SCC, ng/mL ‡	3.6 (0.3–44.3)	3.2 (0.3–44.3)	4.8 (0.5–21.8)	**0.336**
Metastatic lymph nodes at histology	10 (11.4)	10 (20.8)	0 (0.0)	**0.002**

Results are presented as n (%) or median (min-max) as appropriate. Bold font indicates statistically significant values. FIGO: International Federation of Gynecology and Obstetrics. SCC: squamous cell carcinoma antigen. * Partial response includes both microscopic response (21/48) and macroscopic response (27/48). † Grading of differentiation at staging was available in 79 patients. ‡ SCC at staging was available in 73 patients.

**Table 3 cancers-15-03071-t003:** Uni- and multivariable analysis of predictive parameters for pathological partial response prediction within the same imaging at each time point or time interval.

Characteristic	Univariable Analysis				Multivariable Analysis			
	OR (95% CI)	*p* Value	AUC (95% CI) of the Model	*p* Value of the Model	OR (95% CI)	*p* Value	AUC (95% CI) of the Model	*p* Value of the Model
**“Baseline” examination**								
US (n = 79)							0.71 (0.60–0.82)	**0.007**
Color score 3 vs. 4 (n = 88)	0.41 (0.17–0.98)	**0.045**	0.61 (0.50–0.71)	**0.040**	0.37 (0.14–0.95)	**0.040**		
VI (n = 80)	0.97 (0.94–0.99)	**0.038**	0.64 (0.51–0.76)	**0.030**	Removed			
VFI (n = 80)	0.93 (0.87–0.99)	**0.023**	0.64 (0.51–0.76)	**0.010**	Removed			
Tumor peak enhancement (n = 86)	1.00 (1.00–1.00)	**0.029**	0.67 (0.56–0.79)	**0.020**	1.00 (1.00–1.00)	**0.040**		
Rise time (n = 86)	1.06 (0.94–1.20)	0.321	0.63 (0.51–0.75)	0.310				
Wash-in rate (n = 86)	1.00 (1.00–1.00)	0.823	0.69 (0.57–0.80)	0.820				
MRI								
No characteristics included in the present study								
PET/CT (n = 88)							0.71 (0.60–0.82)	**0.004**
SUV_max_ (n = 88)	0.90 (0.83–0.98)	**0.016**	0.69 (0.57–0.80)	**0.001**	NIC			
SUV_mean_ (n = 88)	0.83 (0.73–0.95)	**0.008**	0.71 (0.60–0.82)	**0.0001**	0.83 (0.73–0.95)	**0.008**		
**“Early” examination**								
US (n = 74)							0.71 (0.59–0.83)	**0.002**
Maximum tumor diameter, mm (n = 88)	1.05 (1.01–1.10)	**0.012**	0.67 (0.55–0.79)	**0.004**	1.06 (1.02–1.11)	**0.009**		
Tumor volume, cm^3^ (n = 88)	1.02 (0.99–1.04)	0.064	0.65 (0.53–0.76)	**0.010**				
VI (n = 74)	0.97 (0.95–0.99)	**0.026**	0.65 (0.52–0.78)	**0.020**	0.97 (0.95–0.99)	**0.030**		
MRI (n = 88)							0.68 (0.57–0.80)	**0.001**
Maximum tumor diameter, mm (n = 88)	1.05 (1.02–1.09)	**0.005**	0.68 (0.57–0.80)	**0.001**	1.05 (1.02–1.09)	**0.005**		
Tumor volume, cm^3^ (n = 88)	1.05 (1.01–1.09)	**0.017**	0.68 (0.57–0.80)	**0.001**	NIC			
PET/CT (n = 88)							0.69 (0.57–0.80)	**0.010**
MTV (n = 88)	1.03 (1.00–1.06)	**0.024**	0.69 (0.57–0.80)	**0.010**	1.03 (1.00–1.06)	**0.020**		
TLG (n = 88)	1.00 (0.99–1.01)	0.176	0.68 (0.56–0.80)	**0.003**				
**“Final” examination**								
US								
No characteristics included in the present study								
MRI (n = 82)							0.78 (0.68–0.88)	**0.0001**
Maximum tumor diameter, mm (n = 88)	1.12 (1.04–1.19)	**0.001**	0.71 (0.60–0.81)	**0.0001**	1.09 (1.01–1.18)	**0.040**		
Tumor volume, cm^3^ (n = 88)	2.49 (1.04–5.95)	**0.040**	0.72 (0.62–0.83)	**0.0001**	NIC			
Evaluation according to High DWI SI plus ADC_mean_ ≤ 1.1 × 10^−3^ mm^2^/s (n = 82)	7.75 (2.78–21.59)	**<0.0001**	0.73 (0.63–0.82)	**<0.0001**	3.82 (1.20–12.13)	**0.020**		
PET/CT (n = 88)							0.70 (0.59–0.81)	**<0.0001**
SUV_max_ (n = 88)	2.27 (1.27–4.04)	**0.005**	0.70 (0.59–0.81)	**<0.0001**	2.27 (1.27–4.04)	**0.005**		
SUV_mean_ (n = 88)	3.12 (1.36–7.18)	**0.007**	0.68 (0.56–0.79)	**0.0004**	NIC			
TLG (n = 88)	1.04 (0.99–1.09)	0.170	0.64 (0.53–0.76)	**0.020**				
**Δ “baseline”–“early” examination**								
US (n = 85)							0.71 (0.60–0.81)	**0.0004**
ΔMaximum tumor diameter% (n = 88)	0.97 (0.94–0.99)	**0.009**	0.66 (0.54–0.77)	**0.005**	NIC			
ΔTumor volume% (n = 88)	0.98 (0.97–0.99)	**0.002**	0.71 (0.60–0.81)	**0.0004**	0.98 (0.97–0.99)	**0.002**		
ΔTumor peak enhancement % (n = 85)	0.99 (0.99–1.00)	0.086	0.64 (0.53–0.76)	**0.020**				
ΔWash-in rate% (n = 85)	0.99 (0.99–1.00)	0.070	0.67 (0.55–0.79)	**0.004**				
MRI (n = 88)							0.71 (0.60–0.82)	**0.0003**
ΔMaximum tumor diameter% (n = 88)	0.96 (0.94–0.99)	**0.002**	0.70 (0.59–0.81)	**0.0005**	-	-		
ΔTumor volume% (n = 88)	0.97 (0.95–0.99)	**0.002**	0.71 (0.60–0.82)	**0.0003**	0.97 (0.95–0.99)	**0.002**		
PET/CT (n = 88)							0.78 (0.68–0.88)	**<0.0001**
ΔSUV_max_% (n = 88)	0.96 (0.94–0.98)	**0.001**	0.75 (0.64–0.86)	**0.0001**	NIC			
ΔSUV_mean_% (n = 88)	0.96 (0.94–0.98)	**<0.0001**	0.76 (0.65–0.86)	**0.0001**	0.93 (0.89–0.97)	**0.001**		
ΔMTV% (n = 88)	0.99 (0.98–0.99)	**0.022**	0.70 (0.59–0.81)	**0.0008**	0.96 (0.93–0.99)	**0.012**		
ΔTLG% (n = 88)	0.98 (0.97–0.99)	**0.013**	0.74 (0.63–0.84)	**0.0004**	1.06 (1.01–1.11)	**0.019**		
**Δ “baseline”–“final” examination**								
US								
Evaluation not performed								
MRI (n = 82)							0.70 (0.59–0.81)	**<0.0001**
ΔMaximum tumor diameter% (n = 88)	0.95 (0.92–0.98)	**0.001**	0.70 (0.59–0.81)	**0.0002**	0.95 (0.92–0.98)	**0.002**		
ΔTumor volume% (n = 88)	0.75 (0.60–0.93)	**0.009**	0.70 (0.59–0.81)	**0.0001**	NIC			
ΔADC_mean_ % (n = 82)	1.03 (1.01–1.04)	**0.007**	0.67 (0.55–0.80)	**0.003**	1.02 (1.01–1.04)	**0.030**		
PET/CT (n = 88)							0.80 (0.71–0.90)	**<0.0001**
ΔSUV_max_% (n = 88)	0.87 (0.80–0.93)	**<0.0001**	0.80 (0.71–0.90)	**<0.0001**	0.87 (0.80–0.93)	**<0.0001**		
ΔSUV_mean_% (n = 88)	0.89 (0.84–0.95)	**<0.0001**	0.79 (0.70–0.88)	**<0.0001**	NIC			
ΔTLG% (n = 88)	0.95 (0.89–1.02)	0.141	0.68 (0.57–0.80)	0.060				

Bold font indicates statistically significant values. Removed: parameter removed from the full model (stepwise backward method with a significance level for removal pr = 0.1). NIC: not included in the multivariate analysis to avoid collinearity bias. AUC: area under the curve. CI: confidence interval. US: ultrasound. VI: vascularization index. VFI: vascularization flow index. MRI: magnetic resonance imaging. PET/CT: positron emission tomography/computer tomography. SUV: standardized uptake value. MTV: metabolic tumor volume. TLG: total lesion glycolysis. DWI SI: diffusion weighted imaging signal intensity. ADC: apparent diffusion coefficient.

**Table 4 cancers-15-03071-t004:** Statistically significant parameters for multivariable analysis of models including predictive parameters from different imaging to predict pathological partial response at each time point or time interval.

Characteristic †	OR (95% CI)	*p* Value	AUC (95% CI) of the Model	*p* Value of the Model
**“Baseline” examination (n = 79) ***			0.77 (0.66–0.87)	**0.0003**
VFI (US)	0.99 (0.99–0.99)	**0.011**		
SUV_mean_ (PET/CT)	0.80 (0.69–0.93)	**0.004**		
**“Early” examination (n = 74) ‡**			0.73 (0.61–0.84)	**0.008**
VI (US)	0.97 (0.94–0.99)	**0.030**		
**“Final” examination (n = 82) °**			0.81 (0.72–0.90)	**<0.0001**
Evaluation according to high DWI SI and ADC_mean_ ≤ 1.1 × 10^−3^ mm^2^/s (MRI)	4.04 (1.19–13.75)	**0.030**		
SUV_max_ (PET/CT)	2.47 (1.15–5.34)	**0.020**		
**Δ “baseline”–“early” examination (n = 88) §**			0.80 (0.71–0.89)	**<0.0001**
ΔSUV_mean_% (PET/CT)	0.94 (0.90–0.98)	**0.007**		
ΔMTV% (PET/CT)	0.96 (0.93–0.99)	**0.040**		
ΔTLG% (PET/CT)	1.06 (1.01–1.11)	**0.040**		
**Δ “baseline”–“final” examination (n = 82) ¶**			0.84 (0.75–0.93) Ɨ	**<0.0001**
ΔSUV_max_% (PET/CT)	0.88 (0.81–0.96) Ɨ	**0.004** Ɨ		

Bold font indicates statistically significant values. AUC: area under the curve. CI: confidence interval. US: ultrasound. VI: vascularization index. VFI: vascularization flow index. MRI: magnetic resonance imaging. PET/CT: positron emission tomography/computer tomography. SUV: standardized uptake value. MTV: metabolic tumor volume. TLG: total lesion glycolysis. DWI SI: diffusion weighted imaging signal intensity. ADC: apparent diffusion coefficient. † Selected by the stepwise backward procedure with a *p* < 0.05. * The full model for the stepwise backward procedure included the independent variables: Color score (US), VI (US), VFI (US), tumor peak enhancement (US), SUV_mean_ (PET/CT). ‡ The full model for the stepwise backward procedure included the independent variables: maximum tumor diameter (US), VI (US), maximum tumor diameter (MRI), MTV (PET/CT). ° The full model for the stepwise backward procedure included the independent variables: maximum tumor diameter (MRI), evaluation according to high DWI SI and ADC (MRI), SUV_max_ (PET/CT). § The full model for the stepwise backward procedure included the independent variables: ΔTumor volume% (US), ΔTumor volume % (MRI), ΔSUV_mean_% (PET/CT), ΔMTV% (PET/CT), ΔTLG % (PET/CT). ¶ The full model for the stepwise backward procedure included the independent variables: Δ maximum tumor diameter % (MRI), ΔADC_mean_% (MRI), ΔSUV_max_% (PET/CT). Ɨ Results derived from a full model developed on 82 patients that slightly differ from those shown in Table 3 derived from a model developed on 88 patients.

**Table 5 cancers-15-03071-t005:** Cost analysis of the models developed.

Model	Type of Examination and Timing	Cost per Patient
Model 1	Ultrasonography + Color Doppler + PET/CT at “baseline” examination	1165.09 EUR
Model 2	Ultrasonography + Color Doppler at “early” examination	93.44 EUR
Model 3	MRI + PET/CT at “final” examination	1191.73 EUR
Model 4	PET/CT at “baseline” and “early” examination	2143.30 EUR
Model 5	PET/CT at “baseline” and “final” examination	2143.30 EUR

MRI: magnetic resonance imaging. PET/CT: positron emission tomography/computer tomography.

## Data Availability

The datasets generated and/or analyzed during the current study are available from the corresponding author on reasonable request.

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
