# Peer review of "The Role of Multimodal Imaging in Pathological Response Prediction of Locally Advanced Cervical Cancer Patients Treated by Chemoradiation Therapy Followed by Radical Surgery"

_cancers, 2023, doi:10.3390/cancers15123071_

Round 1

Reviewer 1 Report

It is very interesting, but the methodology seems to be problematic. Firstly, how standardised is the timing of postoperative MRI, CT, PET and ultrasound? Post-operatively, there is inflammation, lymphocysts and also haematomas, which can be assessed for a variety of reasons. Also, ultrasound is only assessed in the same person, which we assume has a significant bias. It is a very difficult method and at the moment it seems that only guesswork can be used to make a claim.

Author Response

Point 1 It is very interesting, but the methodology seems to be problematic. Firstly, how standardised is the timing of postoperative MRI, CT, PET and ultrasound? Post-operatively, there is inflammation, lymphocysts and also haematomas, which can be assessed for a variety of reasons.

Response 1 In this study, all imaging examinations (MRI, PET/CT and ultrasound) were pre-operatively performed: about 3 weeks before treatment (“baseline”), after 2 weeks of treatment (“early” evaluation), and 5 weeks after the end of treatment (“final” evaluation). Radical hysterectomy and pelvic (with or without aortic) lymphadenectomy were planned within 6–8 weeks from completion of treatment. In any case, the choice of imaging timing has been described in the Discussion (page 16, lines 16-25).

 Point 2 Also, ultrasound is only assessed in the same person, which we assume has a significant bias. It is a very difficult method and at the moment it seems that only guesswork can be used to make a claim.

Response 2 All ultrasound examinations were performed by the same examiner (A.C.T.), with more than 15 years of experience in gynecologic ultrasound to avoid interobserver variability pag 3, line 35.

Reviewer 2 Report

I suggest that the title be restructured, it was very confusing and uninformative.

The statistical analysis section was very detailed, however it was very confusing, it could be rewritten and summarized.

Author Response

Point 1 I suggest that the title be restructured, it was very confusing and uninformative.

Response 1 Thank you for your suggestion, we changed the title in: “The role of multimodal imaging in pathological response prediction of locally advanced cervical cancer patients treated by chemoradiation therapy followed by radical surgery.”

Point 2 The statistical analysis section was very detailed; however it was very confusing, it could be rewritten and summarized.

Response 2 Thank you for your suggestion, we modified the statistical analysis section to be more clear (from page, line 46 to page 6 line 4).

Reviewer 3 Report

It should be necessary to stress that the report is the reevaluation of presented previously data on PRICE group of patients. There is delay of current report with the  last PRICE study about 5 years! What is the cause of it ? During this delay it appeared changes in staging, diagnostic methods and treatment recomendations.

Author Response

Point 1 It should be necessary to stress that the report is the reevaluation of presented previously data on PRICE group of patients.

Response 1 We are aware that the current patient population entirely overlap with our previous studies, and we specify this consideration both in the introduction (page 2, lines 35-45) and in the discussion sections (page 16, lines 41-51). The current manuscript differs in analytic methods and provides new, additional analyses already planned in the original protocol, which are complementary to those of previous studies. Its originality is in the attempt to merge the results obtained from the evaluation of more than one parameter to improve the predictive performances either using a single imaging method or integrating more than one.

Point 2 There is delay of current report with the last PRICE study about 5 years! What is the cause of it?

Response 2 The last PRICE study paper was published in 2020. The large number of investigated parameters required a skimming process to select those parameters most representative for each imaging examination. Of the overall 95 parameters extracted from the three imaging modalities, only 34 (36%) were eventually considered in the present study. Moreover, our Institution was highly involved in COVID pandemic and our priorities in the last 3 years had substantially changed.

Point 3 During this delay it appeared changes in staging, diagnostic methods and treatment recomendations.

Response 3 We aware that FIGO staging was updated in 2018, but we preferred to not retrospectively revise it. According to the prospective design of the study, we decided to adopt the previous FIGO stage classification as the inclusion period time was 2010-2014.

Round 2

Reviewer 1 Report

Unfortunately, the authors have not replied to the inquiries we raised. The conclusion should be based on an unbiased medical examination. At least only one examiner, even with 15 years of experience, lacks certainty. Please explain the images in detail. In addition, the authors should explain how to distinguish the lesion through different image methodologies.

Author Response

Point1 Unfortunately, the authors have not replied to the inquiries we raised. The conclusion should be based on an unbiased medical examination. At least only one examiner, even with 15 years of experience, lacks certainty.

Response1 Thank you for your comment. We aware that the involvement of only one examiner, which was applied only for ultrasonography, lacks certainty of data. However, when planning the study, we decided to involve one single ultrasound examiner with very high experience due to the peculiarity of image selection and interpretation of a real time technique as well as the additional use of complex diagnostic technique like infusion of SonoVue. We added this sentence in the discussion (pag 3 lines 28-30)

Point 2 Please explain the images in detail.

Response 2 Thank you for your comment. We added details in the legend of figure 2

Point 3 In addition, the authors should explain how to distinguish the lesion through different image methodologies.

Response3 Thank you for your comment. We explained this point in M&M section (pag 17 lines 5-7)